# Acute, Non-Specific Low Back Pain Does Not Impair Isometric Deadlift Force or Electromyographic Excitation: A Cross-Sectional Study

**DOI:** 10.3390/sports10110168

**Published:** 2022-10-31

**Authors:** Matt S. Stock, Megan E. Bodden, Jenna M. Bloch, Karen L. Starnes, Gabriela Rodriguez, Ryan M. Girts

**Affiliations:** 1Neuromuscular Plasticity Laboratory, Institute of Exercise Physiology and Rehabilitation Science, University of Central Florida, Orlando, FL 32816, USA; 2Department of Natural and Health Sciences, Pfeiffer University, Misenheimer, NC 28109, USA

**Keywords:** low back pain, pain, EMG, deadlift, clinical assessment, resistance training

## Abstract

Low back pain (LBP) is a leading cause of disability. The use of deadlift-based assessments in assessing LBP is becoming common in clinical settings, but these concepts have not been well studied. We sought to compare force and muscle excitation during isometric deadlifts in participants suffering from LBP versus asymptomatic controls. We also compared these outcomes for conventional versus hexagonal barbells. Sixteen adults with mild-to-moderate, acute, non-specific LBP and 19 controls performed maximal, isometric deadlifts while standing on a force plate using conventional and hexagonal barbells. Surface electromyographic signals were recorded from the upper trapezius, external oblique, erector spinae, vastus lateralis, and biceps femoris. Normalized peak force and peak rate of force development were similar for those with acute, non-specific LBP and controls. Surface electromyographic excitation was not different between groups, but was higher with the hexagonal barbell for the vastus lateralis and upper trapezius. Both groups felt equally safe and confident. In summary, the presence of acute, non-specific LBP did not impair peak and rapid force or muscle excitation. Hexagonal barbells may optimize knee extensor and trapezius activation. Deadlift-based force assessments appear safe and may be useful in the assessment of functional strength in patients with acute, non-specific LBP.

## 1. Introduction

According to the World Health Organization (WHO), low back pain (LBP) is the leading cause of disability worldwide [1]. LBP severity varies among individuals, but is characterized as discomfort, tension, or stiffness localized below the costal margin and above the inferior gluteal folds, with or without sciatica [2]. The National Institutes of Health defines acute LBP as pain that lasts a few days to a few weeks, and chronic LBP as a pain that lasts ≥ three months [3]. LBP is a widespread, all-encompassing diagnosis that affects multiple dimensions of one’s life and is associated with depression [4,5]. Darzi et al. [6] examined quality of life in patients with LBP using the WHO Quality of Life Instruments, which assesses physical health, psychological health, social relations, and environmental health. They reported that the LBP group had low scores in every domain, leading to decreased overall quality of life [6]. Given how debilitating LBP can be, the scientific community is increasingly recognizing the need for evidence-based treatments and assessments [7].

Resistance training can help decrease pain levels and improve functional status in patients with LBP [8]. A recent meta-analysis by Tataryn et al. [9] concluded that posterior chain resistance training (defined by the authors as “…exercises and movements that targeted muscles located in the thoracic, lumbar and posterior hip regions that were agonists for hip extension, lumbar and/or thoracic extension, shoulder extension, scapular downward rotation, scapular elevation and scapular retraction”), as well as general exercise, improves chronic LBP via decreased pain, decreased levels of disability, and increased muscle strength. However, posterior chain resistance training had significantly greater effects on these domains when compared to general exercise during 12–16-week programs [9]. Therefore, while a variety of exercises may be useful for treating LBP, those that recruit multiple muscles of the posterior chain seem most effective. Deadlifts are a commonly used compound movement that allow participants to sufficiently load large muscle groups for improvements in strength and power [10,11,12,13]. However, there are almost no high-quality studies that exclusively examine the use of deadlifts as a treatment for patients with LBP. Fisher and colleagues [14] recently searched the clinical literature and identified only three intervention studies that utilized deadlifts in the treatment of patients with LBP.

While deadlifting with a conventional barbell is common, concerns about stress on the lumbar spine, particularly when performed unsupervised by patients unfamiliar with the movement pattern, may be contraindicated. An alternative approach may be to use a hexagonal barbell, which may reduce lumbar stress by facilitating a more upright posture [12,13,15]. Additionally, the hexagonal barbell deadlift allows the external load to be more evenly distributed among the lower extremity joints, thus allowing participants to reach higher peak force, power, and velocity [12]. However, these concepts have only been studied in asymptomatic participants. It is worth determining whether objective measures of strength and muscle activity, as well as perceptions, differ with conventional versus hexagonal barbells in a trained population.

While resistance training appears to be effective in treating individuals with LBP, limited data exists on deadlift performance and muscle excitation in these patients [9]. As such, the primary objective of this study was to compare isometric deadlift strength and muscle excitation in participants suffering from acute, non-specific LBP versus asymptomatic controls. We also sought to compare these outcomes for conventional versus hexagonal barbells. Finally, we sought to compare self-perceived confidence, safety, and force when using the two barbells. We hypothesized that the LBP group would demonstrate lower peak and rapid force, along with a pattern of muscle excitation indicative of compensation for acute LBP, such as lower activity for the spinal erectors and hip extensors. We speculated, however, that use of a hexagonal barbell would minimize between-group differences in maximal force and muscle excitation, and that participants suffering from acute, non-specific LBP would report feeling safer, more confident, and stronger with a hexagonal barbell.

## 2. Materials and Methods

### 2.1. Research Design

This study utilized a cross-sectional design with both between (groups) and within (barbells) participant factors. The participants completed a screening questionnaire, Roland Morris Disability Questionnaire (RMDQ), Fear Avoidance Behavior Questionnaire (FABQ), and Numerical Pain Rating Scale (NPRS) over the phone to screen for eligibility. A single 90-min laboratory visit was scheduled at the conclusion of the call. The participants refrained from alcohol consumption for ≥24 h and resistance training for ≥48 h prior to testing. Conditions in the laboratory were kept constant, including the personnel involved in data collection. All participants were aware of the study procedures and signed informed consent documents. The University of Central Florida Institutional Review Board approved this study (ID# STUDY00003020).

### 2.2. Participants

Thirty-five resistance trained adults between the ages of 18–35 years participated in this study, including ten males with LBP, six females with LBP, nine asymptomatic males, and ten asymptomatic females. Participants were recruited through the University of Central Florida Physical Therapy clinic, word of mouth, and social media. For inclusion criteria, participants were required to have engaged in ≥two resistance training sessions per week over the previous one year. Individuals in the LBP group were required to report recently onset and current pain but score ≤5/10 on the NPRS, ≤15/30 on the FABQ-PA, ≤34/66 on the FABQ-W, and ≤14/24 on the RMDQ. For the NPRS, participants were asked to make three pain ratings, corresponding to current, best, and worst pain experienced over the past 24 h. Our exclusion criteria included neuromuscular or metabolic disease, arthritis, and trouble controlling one’s muscles, as well as use of anabolic steroids, muscle relaxants, benzodiazepines, or corticosteroids in the previous year. Individuals with a history of cancer, heart attack, spinal surgery, radiculopathy, herniated or bulging discs, or sciatica were also excluded. A physician provided guidance on the appropriateness of enrollment when deemed medically necessary.

### 2.3. Subjective Assessment of LBP and Physical Disability

All participants completed questionnaires to ensure current pain and safety over the phone, and scores were again verified during the laboratory visit. The FABQ is a 16 question self-reported questionnaire that assesses an individual’s fear-avoidance beliefs in relation to physical activity and work and how it contributes to an individual’s LBP [16]. The RMDQ is a 24-item questionnaire about how LBP affects an individual’s ability to complete functional activities and provides an overall assessment for functional disability due to LBP [17]. The NPRS is an 11-point scale that is score from 0–10 with 0 indicating no pain and 10 indicating the most intense pain imaginable [18].

### 2.4. Maximal Isometric Deadlifts

All testing was conducted while standing on a force plate (AMTI AccuPower™-Optimized [ACP-O]; Advanced Mechanical Technology, Inc., Watertown, MA, USA) inside of a power rack (Dark Horse™ Rack, Sorinex Exercise Equipment, Inc., Lexington, SC, USA). All participants stood barefoot in the middle of the force plate with their feet hip-width apart, which was measured with an anthropometer. Pins were set in the rack below and above the barbells to fix them into place. The barbells were set at 7.62 cm inches below the apex of the patella for each participant. This bar height was chosen following pilot testing as a means of engaging the hip and knee joints while minimizing the likelihood of lumbar flexion. For the conventional barbell, the participants stood as close to the barbell as possible such that it was over the middle of the foot. An alternated grip was used (i.e., one hand pronated, one hand supinated). For the hexagonal barbell (Rogue TB-1 Trap Bar 2.0, Rogue Fitness, HQ, Columbus, OH, USA), the participants stood in the middle of the barbell and utilized a closed grip. Once the participants were warmed up and familiar with the task, they performed three, five-second maximal isometric deadlifts with each barbell. The participants were instructed to pull on the barbell both “hard and fast.” Strong verbal encouragement was provided. Prior to beginning the deadlift, the participants were as still as possible, ensuring a flat baseline force signal. Three minutes of rest were provided between attempts and barbells. Figure 1 shows examples of the experimental setup for both conditions. Isometric force signals were acquired at 1000 Hz and AccuPower Software 4.0 (ACCUPOWER SOLUTIONS, Dickinson, ND, USA) was used to obtain peak force and the peak rate of force development (RFDpeak). RFDpeak was determined as the highest value of the first derivative of the force signal between onset and peak force [19]. Both peak force and RFDpeak were normalized to body mass (N/kg and N/second/kg, respectively). The order in which participants used the conventional and hexagonal barbells was randomized.

### 2.5. sEMG

Bipolar sEMG signals were recorded from the vastus lateralis, biceps femoris, erector spinae, external oblique, and upper trapezius on the right side of the body (Figure 1). For each muscle, the sensors (Trigno™ EMG, Delsys, Inc., Natick, MA, USA; interelectrode distance = 10 mm, 20–450 Hz) were placed in accordance with recommendations from the sEMG for Noninvasive Assessment of Muscles project [20]. The skin over the belly of each muscle was prepared prior to testing by shaving and cleaning with rubbing alcohol. We inspected the signals for low baseline noise prior to data collection, and additional skin preparation was conducted as needed. sEMG sensors were fixed to the surface of the skin via adhesive tape. EMGworks Acquisition software (version 4.7.6, Delsys, Inc., Natick, MA, USA) was used to record the sEMG signals at a sampling rate of 2000 Hz. Following data collection, the sEMG signals were analyzed with EMGworks Analysis software (version 4.7.6, Delsys, Inc., Natick, MA, USA). The root-mean-square value (μV RMS) from the middle three seconds of the pull was utilized for analysis. Figure 2 shows example sEMG signals with use of the hexagonal barbell.

### 2.6. Perceived Confidence, Safety, and Force

After the use of each barbell, participants were given a two-question survey about their perceived confidence and safety (i.e., four total questions), with 0 representing not confident at all and 10 representing the most confidence possible. Participants were also asked to report which barbell they felt like they pulled with more force.

### 2.7. Statistical Analyses

All statistical analyses were performed with JASP software (version 0.16.3, University of Amsterdam, Amsterdam, The Netherlands) [21]. Differences between groups for ordinal level data were examined with Mann–Whitney U tests. Rank-biserial correlations were examined as measures of effect size, with <0.1 considered trivial, 0.1–0.3 considered a small effect, 0.3–0.5 considered a moderate effect, and >0.5 considered a large effect [22]. The normalized peak force and RFDpeak data were examined with separate two-way (group [LBP, control] × barbell [hexagonal, conventional]) mixed factorial analyses of variance (ANOVAs). Differences in sEMG excitation were examined with a three-way (group × barbell × muscle [biceps femoris, erector spinae, vastus lateralis, upper trapezius, external oblique) mixed factorial ANOVA. If the sphericity assumption was violated, Greenhouse-Geisser corrections were applied. Bonferroni-corrected pairwise comparisons were used to further decompose significant interactions and main effects. The partial eta squared statistic (η^2^) was used as a measure of the effect size for each ANOVA, with 0.01, 0.06, and 0.14 representing small, medium, and large effects, respectively [22]. We also examined Cohen’s d statistics when examining pairwise differences for parametric data. Small, medium, and large Cohen’s d corresponded to 0.2, 0.5, and 0.8, respectively [20]. An alpha level of 0.05 was used to determine statistical significance. Univariate scatterplots in Figure 3 were created with template provided by Weisser et al. [23].

## 3. Results

### 3.1. Description of Study Sample: NPRS, RMDQ, and FABQ

Table 1 displays demographic, pain, and disability data for each group. For the LBP group, NPRS current scores ranged from 1–5, which is indicative of mild-to-moderate pain. All asymptomatic participants rated their current pain as ‘0’, but four participants rated their worst pain in the past 24 h > 0 (scores of 1, 1, 2, and 3). All Mann–Whitney U tests were statistically significant (*p* ≤ 0.002) with moderate/large effect sizes (≥0.381), suggesting greater pain and disability for the participants in the LBP group.

### 3.2. Normalized Peak Force and RFDpeak

Figure 3 displays mean and individual participant data. The normalized peak force results from the two-way mixed factorial ANOVA indicated that there was no group × barbell interaction (F = 0.233, *p* = 0.632, ή^2^ = 0.007). In addition, there was no main effect for group (marginal mean ± standard error of the mean [SEM]: LBP = 20.39 ± 0.67 N/kg vs. control = 19.95 ± 0.67 N/kg; F = 0.207, *p* = 0.652, ή^2^ = 0.006) or barbell (conventional = 20.15 ± 0.52 N/kg vs. hexagonal = 20.19 ± 0.52 N/kg; F = 0.010, *p* = 0.921, ή^2^ < 0.001).

The normalized RFDpeak results indicated that there was no group × barbell interaction (F = 0.013, *p* = 0.909, ή^2^ < 0.001). There were also no main effects for group (LBP = 58.13 ± 7.66 N/s/kg vs. control = 60.24 ± 7.66 N/s/kg; F = 0.038, *p* = 0.847, ή^2^ = 0.001) or barbell (conventional = 59.39 ± 5.78 N/s/kg vs. hexagonal = 58.99 ± 5.78 N/s/kg; F = 0.009, *p* = 0.925, ή^2^ < 0.001).

### 3.3. EMG Excitation

For sEMG excitation, there was no group × barbell × muscle interaction (F = 1.75, *p* = 0.177, ή^2^ = 0.050), no group × barbell interaction (F = 0.027, *p* = 0.871, ή^2^ < 0.001), and no group × muscle interaction (F = 1.411, *p* = 0.247, ή^2^ = 0.041). There was, however, a significant barbell × muscle interaction (F = 18.748, *p* < 0.001, ή^2^ = 0.362). sEMG excitation was higher for the hexagonal barbell versus the conventional barbell for the vastus lateralis (*p* < 0.001) and upper trapezius (*p* < 0.001). Figure 4 displays raincloud plots of sEMG excitation for each muscle and barbell.

### 3.4. Perceived Confidence, Safety, and Force

Table 2 shows the results from the surveys following testing. Generally, participants in both groups felt safe and confident, with mean scores ≥ 8/10. Independent of group, 15 (42.9%) participants felt that they were able to pull with more force with the hexagonal barbell, whereas 20 (57.1%) felt that they were able to pull with more force with the conventional barbell. None of Mann–Whitney U tests for perceived confidence, safety, and force were statistically significant (*p* ≥ 0.120) and showed trivial/small effect sizes (≤|0.296|), suggesting that the LBP and control participants had similar perceptions of the two barbells.

## 4. Discussion

Very little is known about the use of deadlifts and LBP, both from intervention and assessment perspectives. The results of the present study demonstrated that those with acute, non-specific LBP had similar absolute and rapid force and sEMG excitation as asymptomatic controls. Furthermore, both groups reported similar levels of perceived safety, confidence, and strength. However, independent of pain levels, we did observe differences in sEMG excitation between the barbells, with vastus lateralis and upper trapezius activity being higher for the hexagonal barbell. Our findings have important implications for understanding different neuromuscular attributes of individuals suffering from acute, non-specific LBP and developing effective treatment protocols consisting of deadlifts.

Our key finding that participants reporting current LBP did not show reductions in force and sEMG excitation relative to a control group was unexpected. We also did not observe differences between-group differences in sEMG excitation, suggesting that those with acute, non-specific LBP did not adopt a compensatory neuromuscular pattern that would allow them to produce more force. As we are unaware of previous literature addressing this issue, we have little to contrast our findings against. We suspect that an important consideration is the fact that we studied isometric deadlifts at a specified bar height. This approach was more controlled and perhaps conservative than a traditional deadlift, but it may have masked deficiencies that would have been observed had we studied a full range of motion. Future studies are needed to determine the effectiveness, safety, and feasibility of using barbell deadlifts dynamically through a full range of motion in patients suffering from LBP.

Our results add to a growing body of literature comparing conventional and hexagonal barbells [12,13,15,24,25]. While we did not observe differences between groups for any of our dependent variables, we found higher sEMG excitation for the vastus lateralis and upper trapezius with the hexagonal barbell. This finding is consistent with previous studies examining the concentric portion of the range of motion [10,11], and suggests that the more upright posture and arm position with the hexagonal barbell results in greater activity of the knee extensors and scapular elevators. However, the trivial, nonsignificant differences between barbells for sEMG excitation of the biceps femoris, erector spinae, and external oblique conflicts with aspects of previous studies. For example, Camara et al. reported increased erector spinae activity during the eccentric portion of the conventional barbell deadlift compared to the hexagonal barbell [10]. Similarly, Anderson et al. reported 28% greater sEMG muscle excitation for the biceps femoris with a conventional barbell compared to a hexagonal barbell [15]. Overall, while those with acute, non-specific LBP and asymptomatic controls showed no differences in force output and perceptions of the two barbells, the higher level of muscle excitation suggests that use of the hexagonal barbell may result in favorable outcomes compared to a conventional barbell, though much more research is needed.

The present study had four important limitations. First, the sEMG excitation values were not normalized to maximal values, mostly because of time constraints. However, given the homogeneity of the groups because of our inclusion/exclusion criteria and the trivial effect sizes for the differences between groups, we believe it is unlikely that sEMG normalization would have impacted our findings. Second, our ratio of males/females in our LBP group was not even. This limitation was combated by normalizing the force data to body mass. Third, we used a homogenous sample of resistance trained individuals, which may not accurately represent the general population. However, due to the novelty of this research and task specificity, we felt it necessary to study a sample familiar with resistance training. Finally, our study population was limited to those with acute LBP, suggesting that these findings may not be applicable to those with chronic LBP.

In conclusion, contrary to our hypotheses, we observed no differences in absolute and rapid deadlift force and sEMG excitation in participants suffering from acute, non-specific LBP versus asymptomatic controls. We did, however, observe increased sEMG activity for the vastus lateralis and upper trapezius with a hexagonal barbell. Importantly, those suffering from acute, non-specific LBP reported feeling safe and confident during the deadlift assessments, independent of the barbell used. We encourage future investigators to continuing exploring the extent to which deadlifts (and deadlift variations) can be useful in the assessment, prevention, and treatment of LBP.

## Figures and Tables

**Figure 1 sports-10-00168-f001:**
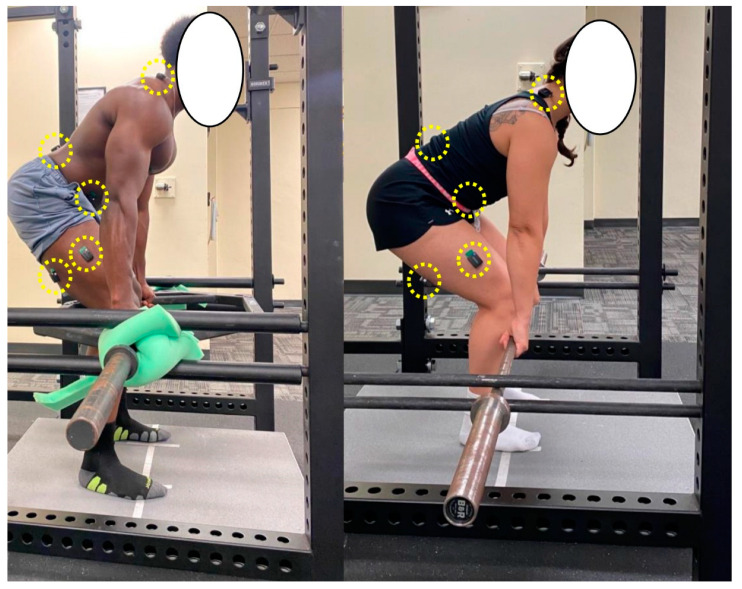
Example positioning on the force plate for both hexagonal barbell (**left**) and conventional (**right**) barbell isometric deadlift pulls. The sEMG sensors have been circled on each participant. Note that for the hexagonal barbell setup, a foam pad was placed underneath the barbell to remove the slack prior to the pull and keep the barbell in the vertical plane.

**Figure 2 sports-10-00168-f002:**
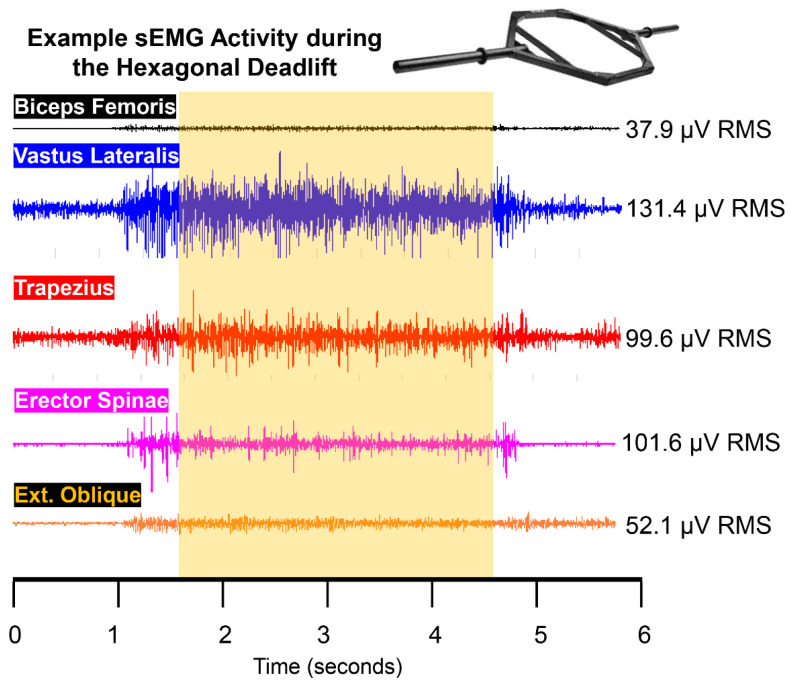
Example sEMG signals for each of the five muscles during an isometric deadlift with the hexagonal barbell. The sEMG excitation values corresponding to the middle three seconds of the attempt are shown to the right.

**Figure 3 sports-10-00168-f003:**
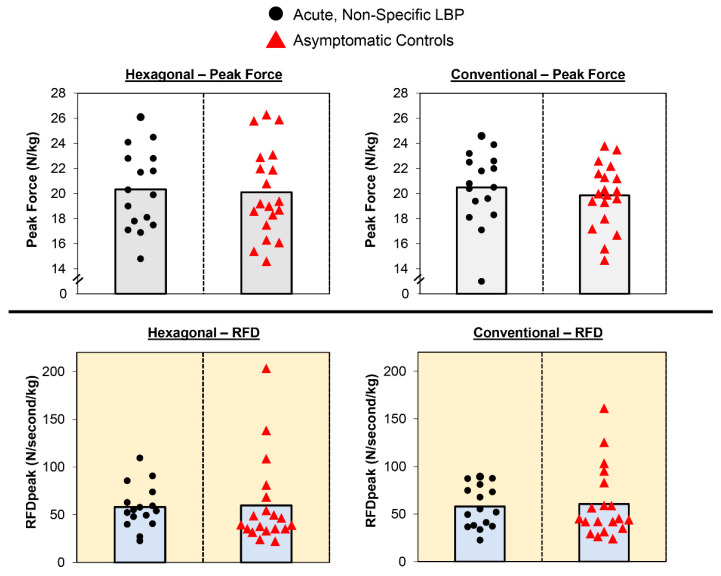
Univariate scatterplots showing individual participant data and means (bar graphs) for normalized peak force (**top**) and RFDpeak (**bottom**).

**Figure 4 sports-10-00168-f004:**
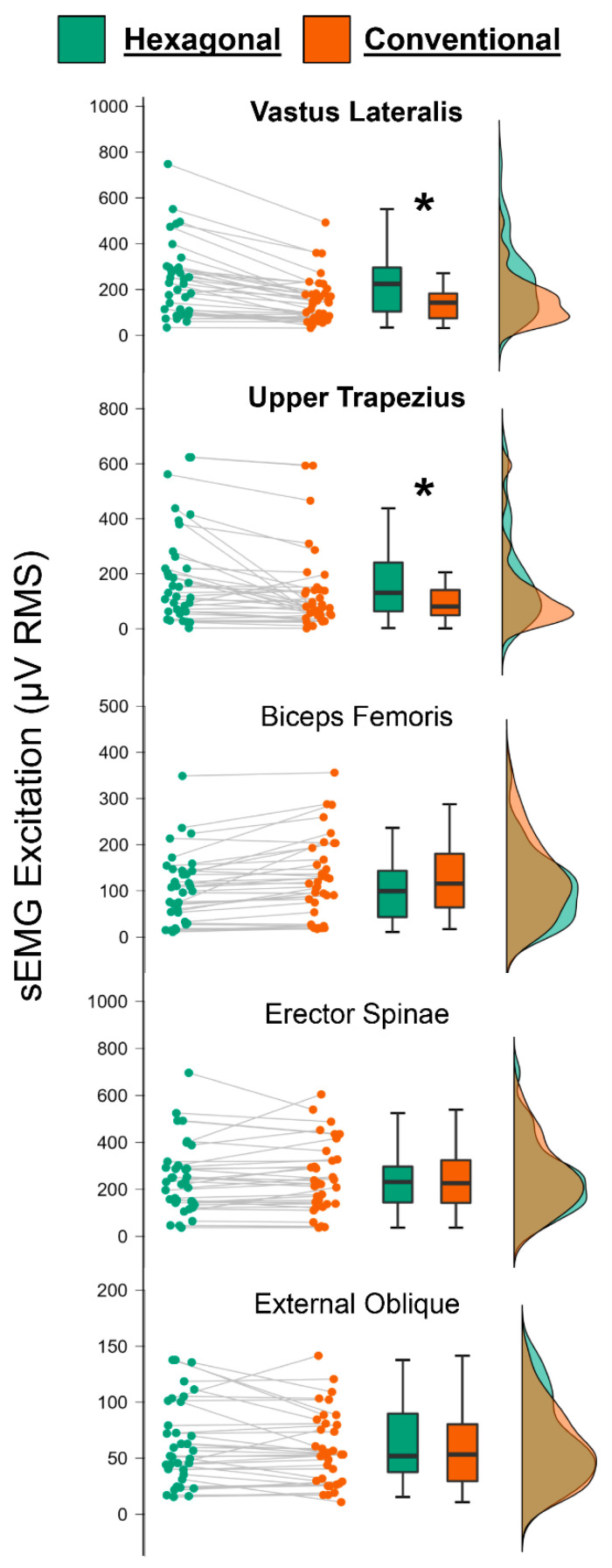
JASP raincloud plots showing sEMG excitation for each of the five muscles using both barbells. For this figure, data has been collapsed across group to show the barbell × muscle interaction. * = statistically significant difference between barbells.

**Table 1 sports-10-00168-t001:** Demographic, pain, and disability data for both groups. Values are not shown for NPRS Current or NPRS Best for the asymptomatic controls, as all participants provided a score of ‘0’. N/A is indicative of when statistical comparisons were not made due to scores of ‘0’ in the control group.

	Acute, Non−Specific LBP	Asymptomatic Controls			
	Mean ± SD (Range)		Mean Difference (95% CI)	*p*	d
Age (years)	22 ± 3 (18–30)	24 ± 4 (18–34)	−1.9 (−4.5–0.7)	0.153	0.497
Height (cm)	172.1 ± 9.7 (152.7–186.2)	171.2 ± 10.5 (148.5–188.0)	0.78 (−6.20–7.73)	0.823	0.077
Mass (kg)	76.4 ± 17.4 (53.2–127.5)	70.6 ± 14.0 (51.2–96.5)	5.81 (−4.99–16.60)	0.309	0.371
BMI (kg/m^2^)	25.7 ± 4.3 (18.9–36.8)	24.0 ± 3.2 (17.7–30.0)	1.721 (−0.87–4.31)	0.185	0.459
	Mean ± SD		Hodges−Lehmann Estimate (95% CI)	*p*	Rank biserial correlation
NPRS Current	2.00 ± 0.97 (1–5)	0 ± 0	N/A	N/A	N/A
NPRS Best	0.56 ± 0.96 (0–3)	0 ± 0	N/A	N/A	N/A
NPRS Worst	3.44 ± 1.99 (1–9)	0.53 ± 1.02 (0–3)	3.0 (2.0–3.0)	<0.001	0.865
RMDQ	1.88 ± 2.09 (0–7)	0.16 ± 0.37 (0–1)	1.0 (0.0–2.0)	0.002	0.546
FABQ Physical Activity	7.44 ± 3.72 (1–13)	0.79 ± 1.81 (0–6)	6.8 (4.0–9.0)	<0.001	0.914
FABQ Work	4.38 ± 5.45 (0–14)	1.16 ± 3.45 (0–16)	0.0 (0.0–8.0)	0.017	0.381

**Table 2 sports-10-00168-t002:** Responses to Likert scale questions concerning perceived safety, confidence, and force production. Data provided in the second and third columns correspond to mean ± SD (range).

	Acute, Non-Specific LBP	Asymptomatic Controls	*p*	Rank Biserial Correlation
“Did you feel safe pulling with the hexagonal barbell?”	9.5 ± 1.1 (5–10)	9.7 ± 0.5 (9–10)	0.914	0.020
“Did you feel safe pulling with the conventional barbell?”	9.1 ± 1.6 (4–10)	9.6 ± 0.8 (7–10)	0.443	0.128
“Did you feel confident pulling with the hexagonal barbell?”	8.3 ± 1.5 (5–10)	8.7 ± 1.4 (5–10)	0.428	0.155
“Did you feel confident pulling with the conventional barbell?”	8.3 ± 1.7 (6–10)	9.2 ± 1.0 (6–10)	0.120	0.296
“Which barbell do you believe you pulled with more force?” 0 = hexagonal, 1 = conventional	0.63 ± 0.50	0.53 ± 0.51	0.576	0.099

## Data Availability

Data will be shared upon email request from the corresponding author (matt.stock@ucf.edu).

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
