# Peer review of "Acute, Non-Specific Low Back Pain Does Not Impair Isometric Deadlift Force or Electromyographic Excitation: A Cross-Sectional Study"

_sports, 2022, doi:10.3390/sports10110168_

Round 1

Reviewer 1 Report

First of all, I would like to thank the authors for their contribution to the scientific field.

I leave you some minor corrections that in my humble opinion will improve this good work.

Abstract:

-I would recommend you not to include abbreviations in the abstract.

Introduction:

-I understand that due to the specificity of the journal and the background that the authors have, they understand the concept of "posterior chain", but a small definition would be appropriate to facilitate the understanding of readers who do not belong to the area of sports science.

Material and Methods:

-Likewise, I would recommend that you combine the exclusion criteria (lines 96 to 99). So that they are easier to understand and do not appear as separate sentences.

-It would also include some citations from previous literature about the measurement protocol developed in the study. Issues such as the location of the bar with respect to the patella or the rest periods (although I understand they are full recovery)

Results:

-The title of Figure 3 has an editorial error, I understand that the peak force is at the top and the RFD is at the bottom.

References:

-Complete citations 3 and 19 with more data fields to make them findable and fit the format.

I strongly encourage you to continue with this line of research so interesting for the training world, recruiting a larger sample and including the full range of motion involved in exercise.

Best of luck in your work and a pleasure!

Reviewer 2 Report

 I congratulate the authors on an ambitious cross-sectional study. The research is robust and the design well considered. I look forward to seeing the end result of this work when it is finally complete and published. I commend the authors for their work - both all of the work leading up to this point and for the planning of this investigation - their contribution to the low back pain literature. I do have some comments about certain methodological issues covered below under MAJOR ISSUES the majority of which are related to clarity of the work as it is currently written.

TITLE

The title should be amended slightly to ensure that the reader understands the type of research immediately that this paper for clarity, interes and ease of read.

ABSTRACT

It is hard to get the detail in an abstract when the word count is limited and this is often the hardest part of a paper to write. However, I do feel that it would be beneficial to explain what specifically you are looking at in relation to low back pain  (this also applies to the main body of the paper). Is it the development of muscle force with low back pain . This needs to be made clearer throughout the paper

KEYWORDS:

Please use recognised MeSH terms as this will assist others when they are searching for information on your research topic. The following website will provide these (simply start typing in a keyword and see if it exists or find an alternative if it does not): https://www.ncbi.nlm.nih.gov/mesh

The introduction is weak. An introduction should announce your topic, provide context and a rationale for your work, while catching the reader´s interest and attention. The above has not been given in the introduction that I have read. Thus, I suggest in this section should be improved, with more details about prevalence, impact related with  low back pain. It is indeed important paper but it lacks several critical references, in which it was presented related with this condition, and it should be emphasized in the INTRODUCTION or Discussion of the authors' paper. More info info in:

Evaluation of Depression in Subacute Low Back Pain: A Case Control Study.

https://www.ncbi.nlm.nih.gov/pubmed/28535558

Relationship of depression in participants with nonspecific acute or subacute low back pain and no-pain by age distribution.

https://www.ncbi.nlm.nih.gov/pubmed/28138263

Also, please describe the hypothesis in this section.

MATERIAL AND METHODS:

This section are appropriate and described in adequate detail while the conclusions clearly link to the data presented. Please, expand and clarification information related with this research for adhere to reporting STROBE guidelines.

RESULTS:

The results section is very appropriate according to the developed methods and the journal´s scope.

DISCUSSION:

Include this section the principal strengths and weaknesses in relation to other studies, discussing important differences in results; the meaning of the study: possible explanations and implications and unanswered questions and future research

CONCLUSION:

summarize the conclusions in order to reflect only the study findings.

Round 2

Reviewer 2 Report

In their first revision of manuscript, the authors have addressed my questions/comments properly.